

# Spatial patterns and driving factors of plant diversity along the urban–rural gradient in the context of urbanization in Zhengzhou, China

Lingling Zhang[1,2], Chong Du[3], Wenhan Li[1], Yongjiang Liu[1], Ge Zhang[1], Shanshan Xie[1], Yiping Liu[1] and Dezheng Kong[1]

[1] College of Landscape Architecture and Art, Henan Agricultural University, Zhengzhou, Henan Province, China
[2] Department of Art and Design, Zhengzhou Business University, Gongyi, Henan Province, China
[3] SuperMap Software Co., Ltd, Zhengzhou, Henan Province, China

Corresponding author
Dezheng Kong,
kdzhenau@henau.edu.cn

## ABSTRACT

Plant diversity is the basis for human survival and development, directly affecting the function and stability of urban ecosystems. Its distribution pattern and causes have been a central issue in ecological and landscape gardening research. Rapid urbanization in Zhengzhou City has led to the fragmentation of urban green spaces and damage to ecosystems, seriously affecting urban biodiversity conservation. Understanding the distribution pattern of plant diversity in the region and its relationship with environmental factors is crucial for maintaining and enhancing urban plant diversity. Plant data from 178 sample plots in the built-up area of Zhengzhou City were collected and combined with environmental factors, and the characteristics of plant diversity, richness patterns, and their main environmental explanations in Zhengzhou City were explored. Results showed that there were 596 plant species belonging to 357 genera and 110 families in the study area. There were five dominant families and four dominant genera. Four distinct spatial patterns of plant diversity were identified along the urban–rural gradient. Urbanization factors such as GDP *per capita*, house prices, and imperviousness within 500 m from the patch significantly influenced plant diversity. There was an imbalance between the spatial pattern of plant diversity and application of urban landscape greening in Zhengzhou City. Future studies should focus on the application of native plants, curb plant homogenization, and reduce anthropogenic interference, which are conducive to protecting and enhancing urban plant diversity. These results can provide a basis for understanding the distribution pattern and influence mechanism of urbanization factors on plant diversity and serve as a reference for policymakers and planners of plant diversity conservation in Zhengzhou City.

## INTRODUCTION

In recent decades, urban expansion has seriously impacted biodiversity and human well-being. According to a United Nations report, by 2050, the global population will reach

9.7 billion, and 68% of the population will live in urban areas (*United Nations, 2017*). Historically, urbanization has been one of the main causes of natural habitat loss. Between 1992 and 2000, urban growth resulted in the loss of 190,000 km$^2$ of natural habitat globally, accounting for 16% of the total global loss of natural habitat during this period. By 2030, urban growth will threaten 290,000 km$^2$ area of natural habitats (*The Nature Conservancy, 2018*). Urban green space plays an important role in improving the urban environment while maintaining the balance of the urban ecosystem (*Loke & Chisholm, 2022*; *Redon et al., 2014*). Although artificial green spaces, such as urban gardens, parks, and green belts, promote urbanization to a certain extent (*Hu et al., 2022*), their patchy distribution and fragmented habitats influence the distribution and diversity of urban plants, thereby affecting the enhancement of urban green space system functions and development (*Jentsch et al., 2012*). Plant diversity is the mainstay of biodiversity, which guarantees the maintenance of ecosystem services and functions and improves human well-being (*Adhikari & Hansen, 2018*; *Egoh et al., 2020*; *Fan et al., 2017*). However, rapid urbanization poses a serious threat to biodiversity, with the maintenance and protection of urban plant diversity facing significant challenges (*Filazzola, Shrestha & MacIvor, 2019*; *Piana et al., 2019*; *Singh, Singh & Singh, 2018*).

With the dramatic loss of global biodiversity (*Chapin et al., 2000*), researchers are increasingly focusing on the impact of urbanization on plant diversity. The urban–rural gradient method is effective for estimating the level of anthropogenic disturbance in cities (*Burton, Samuelson & Mackenzie, 2009*; *Qi et al., 2024*). This level is expressed as a change from the urban center to the outer edges of the city. It is widely used to measure changes in physical variables, habitat types, and biodiversity in towns and cities (*Crawford & Rudgers, 2012*; *Wang et al., 2020b*). Spatial distribution patterns of different types of plants vary across urban–rural gradients. A previous study showed that the richness of exotic plants in rural areas is lower than that in urban areas (*Vakhlamova et al., 2014*). The species richness of all plants and native plants were the lowest in the city center (*Ranta & Viljanen, 2011*; *von der Lippe & Kowarik, 2008*). In some cities, herbaceous plant diversity indices showed higher performance in suburban areas (*Aronson et al., 2015*). Plant distribution patterns are influenced not only by natural environmental factors but also by urbanization factors (*Siles, Voirin & Benie, 2018*; *Ye et al., 2020*). *Luo et al. (2007)* indicated that the early spring phenology of vegetation in eastern North America was considerably affected by urban warming, demonstrating that the urban heat island effect accelerates the spring recovery of vegetation. *Zhu et al. (2024)* reported that urbanization of the Yangtze River Delta in China resulted in an earlier start and delayed end of the vegetation growing season, effectively extending the vegetation growing cycle. *Jeong et al. (2019)* revealed that increased population density in Seoul, Korea, contributed to the early start and delayed end of the vegetation growing season. Many studies have focused on anthropogenic factors (*Cheng et al., 2022*; *Guo et al., 2018*; *Xu et al., 2019*). According to *Liu et al. (2023)*, increased attention has been paid to the value of ornamental species, especially the introduction of more exotic plants to botanical gardens, which has led to plant homogenization, and the risk of biological invasion, which has reduced the diversity of associated plants and animals

such as birds to a certain extent. Urban greening is mostly dominated by non-native plants, and it is particularly important to improve native biodiversity in the management and planning of urban spaces (*Nizamani et al., 2021*).

Despite widespread concern that plant diversity is affected by urbanization, no consistent conclusions have been reached on the trends in plant diversity along the urban–rural gradient. Although some studies on urbanization and plant diversity have been conducted in China, their study areas mainly included cities or regions with higher economic levels (*Chapman et al., 2017*; *Du et al., 2016*; *Zhao et al., 2014*). Studies on the impact of urbanization on plant diversity have been neglected in central China. For example, Zhengzhou, the capital of Henan Province and the core city of the Central Plains Economic Zone, has experienced rapid urbanization over the past three decades, driving regional and localized environmental changes in land-use practices, hydrology, and biochemical attributes. Therefore, it is a precise location to explore the link between urbanization and plant diversity in central China (*Zhao & Miao, 2022*). Exploring the distribution pattern of urban plant diversity and its influencing factors can provide a more theoretical and practical basis for the scientific application of urban plants. The results can propose effective solutions to existing problems, provide a reference for the development of species conservation planning, and maintain and improve urban plant diversity (*Chen et al., 2011*).

The changes in plant species diversity driven by urbanization are mainly based on theoretical studies, lacking quantitative analyses (*Hanspach et al., 2016*). The specific objectives of this study were achieved based on the following questions: (1) What is the distribution pattern of species richness of different types of plants along the urban–rural gradient in the Central Plains cities? (2) What are the main factors influencing the species richness of different types of plants? (3) What are the differences in the relative contributions of major influencing factors? (4) What are the strategies for maintaining and enhancing urban plant diversity?

## MATERIALS AND METHODS

### Study area

Zhengzhou City is situated in the flatlands of the middle and lower sections of the Yellow River in the central northern part of Henan Province. Geographically, it lies between 112°42′E–114°14′E longitude and 34°16′N–34°58′N latitude (Fig. 1). It is a typical mega-city in China, with an area of 7,567 km² and a population of 12.74 million (*Zhengzhou Bureau of Statistics, 2022*). The climate is rainy and hot at the same time and dry and cold during the same season. The annual temperature and precipitation are 14.4 °C and 652.9 mm, respectively, and the frost-free period is 220 days. Zhengzhou, the site of the Shang Dynasty capital, was selected as one of the eight ancient capitals in 2004 and has gradually become an economically developed and densely populated city in central China over the years. Its favorable geographical environment has nurtured rich plant and animal resources; however, the rapid development of the urban area is challenging the

![PeerJ]

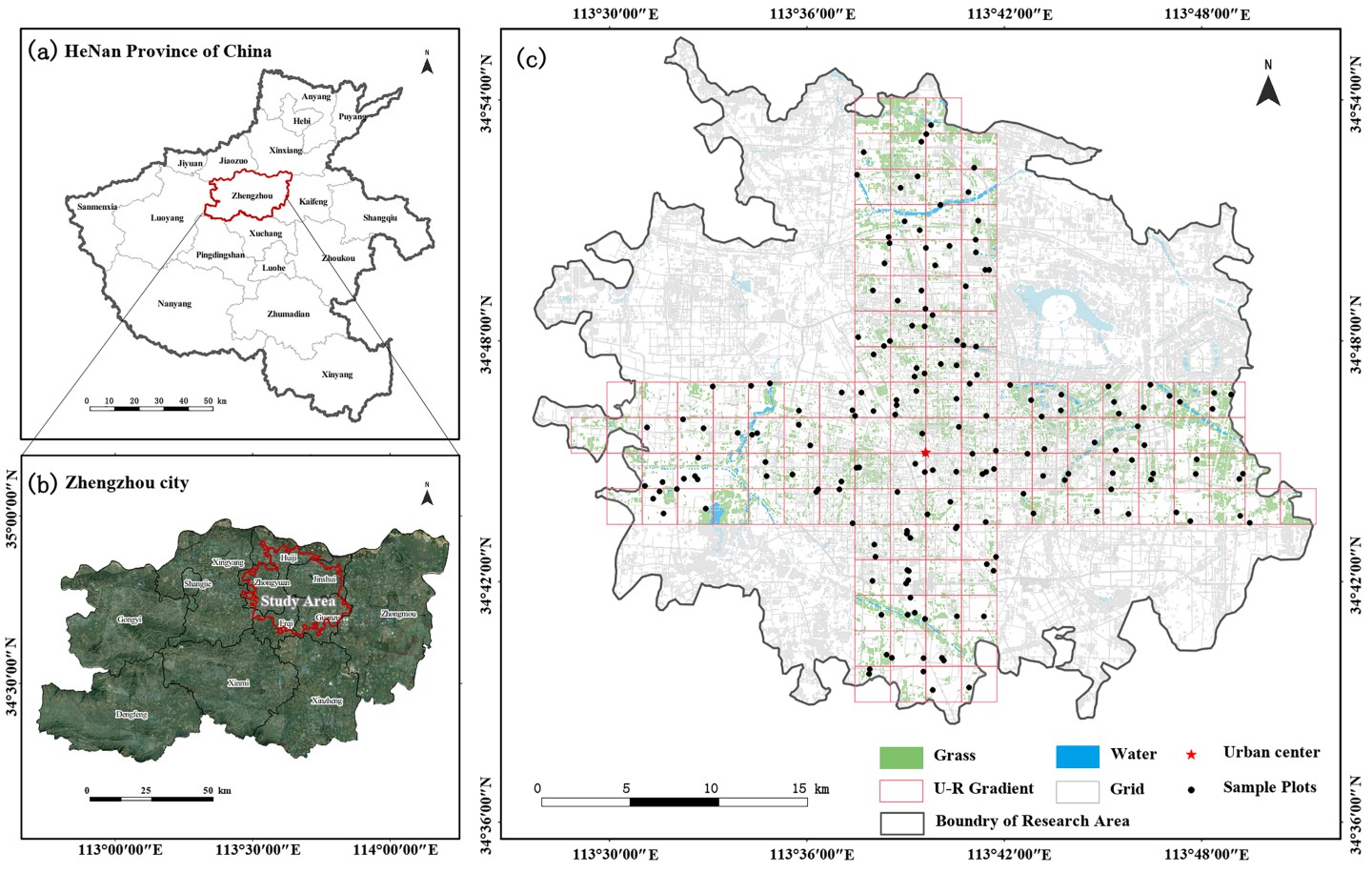

**Figure 1** The location of Henan Province and Zhengzhou City in Central China (A and B); study area, urban–rural transects and the distribution of sampling plots (C).

limits of the ecological environment, endangering the health of urban residents and sustainable development of the region (*Scherer-Lorenzen et al., 2022*). Therefore, this study focused on the built-up area of the city, which is located in the center of Zhengzhou, with a total area of 376.77 km² and urban population of about ~5.016 million people. According to the statistics, until 2022, the green space rate in Zhengzhou City is 36.81% (*Zhengzhou News, 2023*). It is expected that by the end of the 14th Five-Year Plan, the green space rate of the built-up area will reach 37.5% (*ZY News, 2021*).

## Sampling design and community surveys

To obtain spatially representative samples, a checkerboard random-sample layout method was used (*Zhao et al., 2014*). Using a 10-m resolution land-use cover map, green space cover data regarding three types of forests, shrubs, and grasslands were extracted, and the built-up area of Zhengzhou City was divided into a 2 × 2 km grid. Two 8-km wide cross-sections were established along the two longest axes in the city center, namely, east–west (W–E) and north–south (S–N) axes (*Qi et al., 2024*; *Wang et al., 2020b*), with lengths of 40 and 34 km, respectively. The main types of urban green space are park green

space, protective green space, square land, ancillary green space, and regional green space (*Ministry of Housing and Urban-Rural Development of China, 2017*). Based on the proportion of different green space types, stratified sampling for different green space types was conducted, and the number of samples was determined based on the area of each green space type. One to three large green patches were randomly selected in each grid, and the sample area was uniformly set to 400 m². Priority was given to square sample plots of 20 × 20 m; however, if conditions did not allow, adjustments were made to obtain rectangular sample plots. China's current biodiversity conservation planning mainly focuses on green space system planning at the stage of urban master planning and plant diversity conservation planning (*Gan & Wu, 2018*). Therefore, this study mainly focused on the plant diversity of urban green space. For inaccessible sampling sites without vegetation, another similar site within the 500-m radius was searched as a replacement, and a total of 178 sample plots were finally identified (Fig. 1).

From March to October 2023, field investigations were conducted in each sample plot using tools such as portable GPS locators, three-dimensional laser rangefinders, breast diameter rulers, flower stems, measuring tapes, and cameras. Relevant information such as survey time, weather, sample number, location, green space type, and specific information on trees, shrubs, and herbaceous plants, were recorded (Table 1). The names of plant species, whether they are native or not, and identification of spontaneous plant species were based on the Flora of China (*Flora of China Editorial Committee, 2007*), the Plant Plus of China (*Plant Intelligence, 2020*), the Plant novae from Henan (*Ding, Wang & Zhao, 1980*), and historical documents. Cultivated and spontaneous plants were preliminarily classified according to the actual situation of the site. The identification of exotic species was based on the Catalogue of Invasive Plants of China (*Shou, Yan & Ma, 2014*). Plant health status was determined by observing plant growth, checking the color and texture of leaves, and assessing leaves for pests and diseases. Using the difference percentage grading method, plant health status was classified into five levels (*Nagase & Dunnett, 2012*), which were healthy (90–100%; flourishing branches and leaves, bright color, no infestation by pests and diseases); good (70–89%; overall good growth, complete leaves, individual leaves with curled edges, infestation by pests and diseases); fair (40–69%; small areas of withered leaves, small number of branches and leaves infested by pests and diseases); poor (10–39%; large areas of withered leaves, most of the branches and leaves infested by pests and diseases), and very poor (0–9%; dead or severely infested).

## Plant diversity calculations

The raw data obtained from the survey were classified according to the sample plots into three categories: trees, shrubs and herbs. The relative plurality, relative frequency, relative significance (trees) and relative cover (shrubs and herbs), relative importance value (IV), and species diversity index of the species were calculated. IV was used to screen for dominant plants in the community. In general, the greater the IV of a particular plant, the greater the dominance of that plant in the sample site (*Wang et al., 2019*). The calculation method is shown in Eq. (2-1).

 

**Table 1 Targets and contents of the survey.**

| Targets | Content of the survey | Description of content |
|---|---|---|
| Tree layer | Name | Name of each tree species. |
| | Number | Total number of each tree species in the sample. |
| | Diameter of chest | Diameter at breast height of each tree in the sample plot at 1.2 m above ground level. |
| | Crown width | Width of each tree in the sample in the east-west and north-south directions. |
| Shrub layer | Name | Name of each shrub. |
| | Area | If the shrubs in the sample are planted individually, the area is the actual footprint of each plant. If planted in patches, the area is the overall footprint of the plants planted in each patch. |
| | Number | The number of shrubs in the sample is the actual number of plants if they are planted as individual plants. If the shrubs are planted in patches, the number of plants is calculated as the number of plants in 1 square metre and multiplied by the planted area to obtain the total number of plants. |
| | Cover | Ratio of the area of the vertical projection of the above-ground part of the plant to the area of the sample plot. |
| | Height | For shrubs in solitary form, the height is the actual height of each plant. For shrubs planted in patches, the height is the average height. |
| | Width | For shrubs in solitary form, the width is the actual width of each plant. For shrubs planted in patches, the width is the average width. |
| Herbaceous layer | Name | Name of each herb. |
| | Area | The sum of the area occupied by each plant in the sample plot. |
| | Height | Mean height of multiple individuals of each plant within the sample. |
| | Cover | Ratio of the area of the vertical projection of the above-ground part of the plant to the area of the sample plot. |

Importance value formula

$$IV = \frac{Dr + Fr + Pr}{3}.$$ (2-1)

In the formula, $Dr$ represents the relative multiplicity, $Fr$ is the relative frequency, and $Pr$ is the relative significance or coverage.

Species richness is generally expressed in terms of the number of species in the region and calculated using the Margalef species richness index ($R$). Species diversity index is a comprehensive quantitative index reflecting the number, structure, and evenness of distribution of species, and Shannon–Wiener species diversity index ($H$), Simpson species dominance index ($D$), and Pielou's species evenness index ($J$) are generally used (*Benayas & Scheiner, 2002*; *Li et al., 2020*). The calculation methods for each index are as follows:

Margalef species richness index

$$R = \frac{S-1}{l_n N}.$$ (2-2)

Shannon-Wiener species diversity index

$$H = -\sum_{i=1}^{S} (P_i)(l_n p_i).$$ (2-3)

Simpson species dominance index

$$D = 1 - \sum_{i=1}^{S} (P_i)^2. \tag{2-4}$$

Pielou's species evenness index

$$J = \frac{H}{l_n S}. \tag{2-5}$$

In the formula, $S$ is the number of all plant species in the sample plot, $N$ is the total number of individuals of all species, $P_i$ is the proportion of individuals of species i relative to the total number of species in the sample plots, and $P_i = N_i/N$, $N_i$ is the number of individuals of the species i (*Chesson, 2018*).

## Delineation of the urban–rural gradient

*McDonnell & Hahs (2008)* analyzed the urban–rural gradient. Many scholars often use the method when studying plant distribution, landscape patterns, and ecosystem characteristics affected by urbanization (*Qi et al., 2024*). In the current study, the gradient analysis method (Fig. 2) was used to mark the grid at Zhengzhou City center as 0, with positive values in the grid toward north and east directions and negative values in the grid toward south and west directions. Two transects along the W–E and S–N axes were divided into four urban–rural sample strips, and the distribution patterns of plant species diversity in urban–rural sample strips in different directions were analyzed.

## Statistical analysis methods

Sample data were collated and calculated using Excel. The differences and characteristics of plant species in four transects were compared using Venn plots and principal coordinate analysis (PCoA) coordinates, and the differences in plant diversity indices were analyzed using radar plots. Based on the normality test of the dependent variable data, regression analysis was used to fit the optimal model to the mean value of plant diversity on the gradient. Polynomial regression equations, such as linear regression and quadratic quintic term, were compared to screen the optimal model. The criterion was that the $p$-values of the model and parameters were significant, and the larger the $R^2$ the better, the smaller the Akaike information criterion the better (*Wang et al., 2020b*). The regression equation is as follows:

Univariate linear regression equation

$$Y = \beta_0 + \beta_1 x + \varepsilon. \tag{2-6}$$

Polynomial regression equation

$$Y = \beta_0 + \beta_1 x + \beta_2 x^2 + \cdots + \beta_n x^n + \varepsilon. \tag{2-7}$$

In the formula: $Y$ is the dependent variable, which is the variable we hope to predict or explain; $x$ is the independent variable, used to explain the dependent variable; $\beta_0$ is the intercept, which represents the predicted value of the dependent variable when the independent variable is set to 0; $\beta_1, \beta_2, \ldots \beta_n$ are the coefficients, which correspond to the

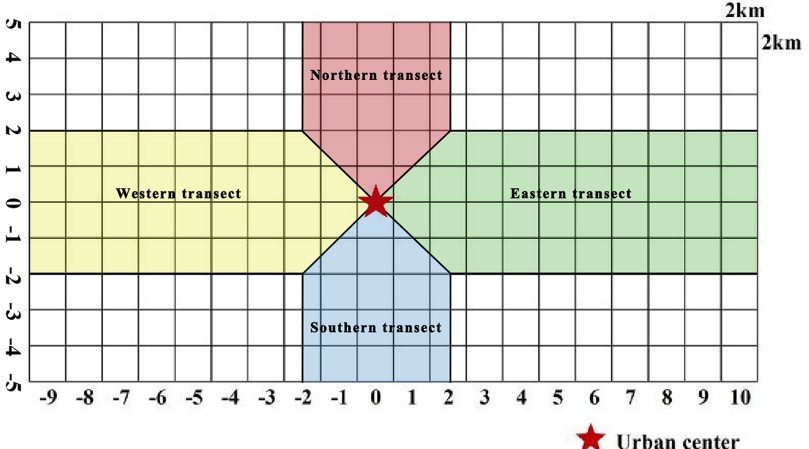

**Figure 2 Schematic diagram of urban–rural gradient division.**

coefficients of the first, second to nth-order terms of the independent variable, denotes the magnitude of the effect of the dependent variable by each power of the independent variable; and $x^2, x^3, \ldots x^n$ are the second, third to the nth powers of the independent variable used to construct a polynomial regression model. The error term $\varepsilon$ represents the difference between the value of the dependent variable and the regression equation.

A generalized linear model was used to analyze the factors affecting plant diversity in different plant types (Table 2), the richness of plant species in different plant types was used as the dependent variable of the model, and the urbanization characteristic factors were used as the independent variable. The independent variable was standardized and centralized to make the data conform to a normal distribution. Pairwise correlation testing on variables was performed using Spearman correlation analysis. If the correlation coefficient between variables was >0.7, only the variable with the largest correlation coefficient with the independent variable was retained in the model; after the model was constructed, the variance inflation factor (VIF) of the model was further examined, and the variables with VIF > 5 were removed from the model again one by one until all variables with VIF < 5 (*Aspinall, 2002*; *Gao et al., 2023*). To clarify the individual explanatory rates of each variable in the model, this study decomposed $R^2$ to obtain the relative contribution of each independent variable.

All of the above calculations were performed in R version 3.4.1 (*R Core Team, 2017*). In this case, data were normalized using the scale function in the base package. The Spearman correlation test used the corrplot function in the corrplot package. The model collinearity test used the check multicollinearity function in the performance package. The model fitting used the glmer function from the lme4 package. Model selection and model averaging used the dredge and model.avg functions. Finally, the calculation method of the r.squared GLMM function in the MuMIN package was used to decompose $R^2$. The figure was completed using the "ggplot" software package in R and Origin 2024.

**Table 2  The overview of models' variables.**

| Variable category | Variable content | Variable abbreviations | Data source |
|---|---|---|---|
| Dependent variable | All plant richness | APR | Calculated acquisition |
| | Tree plant richness | TPR | |
| | Shrub plant richness | SSR | |
| | Herbaceous plant richness | HSR | |
| | Native plant richness | NPR | |
| | Exotic plant richness | EPR | |
| | Cultivated plant richness | CPR | |
| | Spontaneous plant richness | SPR | |
| Independent variable | Distance from patch to city boundary | BD | Calculated acquisition |
| | Proportion of sealed surface around the patches within 500 m | $Sealed_{500}$ | |
| | Landscape shape index | LSI | |
| | Spatial distribution of the population | SDP | China's resource and environmental sciences data centre (https://www.resdc.cn/) |
| | GDP of individual | GDP | |
| | Prices of house | HP | Statistical yearbook download station (https://www.zgtjnj.org/) |
| | Night light index | NL | Earth observation group (https://eogdata.mines.edu/products/vnl/) |

# RESULTS

## Composition of plant resources

In this survey, there were 596 species of vascular plants on 178 sample plots (Fig. 3) belonging to 110 families and 357 genera. According to plant traits, there were 230 species of woody plants belonging to 65 families and 126 genera, including 140 evergreen species and 90 deciduous species. Among these, there were 106 species of trees (17.79% of all plants) belonging to 37 families and 65 genera. There were 124 species of shrubs (20.81%) belonging to 43 families and 72 genera. There were 366 species of herbaceous plants (61.41%) belonging to 70 families and 244 genera; among these, 138 species were annual herbaceous plants and 228 species were perennial herbs. According to the source of species, there were 146 species of native plants and 450 species of exotic plants, accounting for 24.50% and 75.50% of the total species count, respectively. Based on the form of plant cultivation, there were 416 cultivated plant species and 180 spontaneous plant species, accounting for 69.80% and 30.20% of the total species, respectively. According to the results of families, genera, and species, the plant life forms with the highest number of species in urban green spaces were herbaceous plants, followed by shrubs and trees. Among the surveyed plants, there were three species belonging to the first level of national protection, four species belonging to the second level of national protection, 144 nonhazardous plant species, three endangered plant species, five vulnerable plant species, one relict plant species, six species unique to China, and four key protected tree species in Henan.
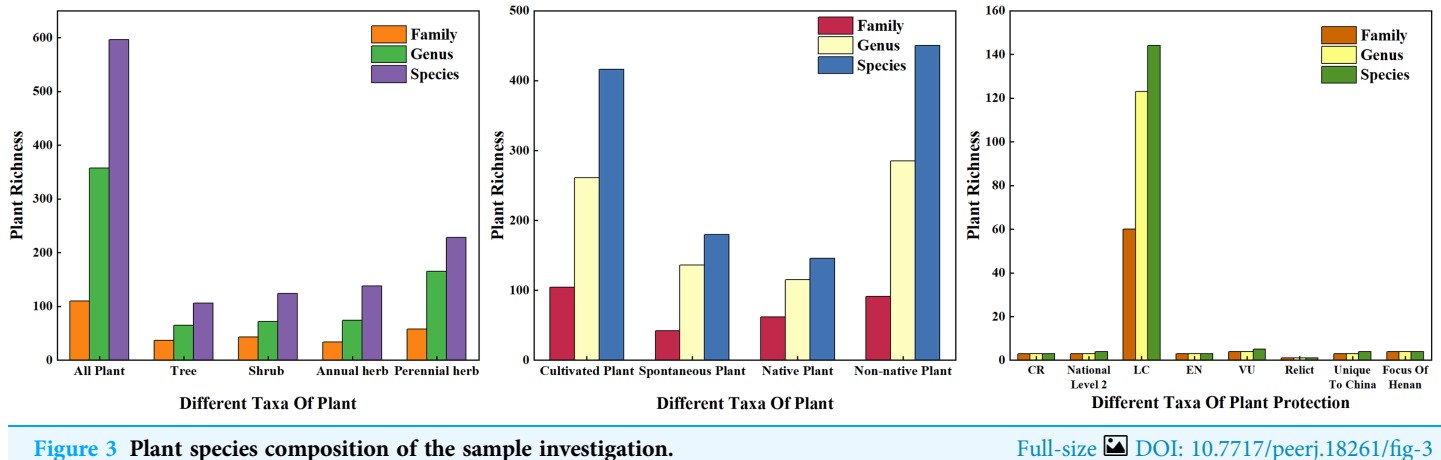

**Figure 3** Plant species composition of the sample investigation.

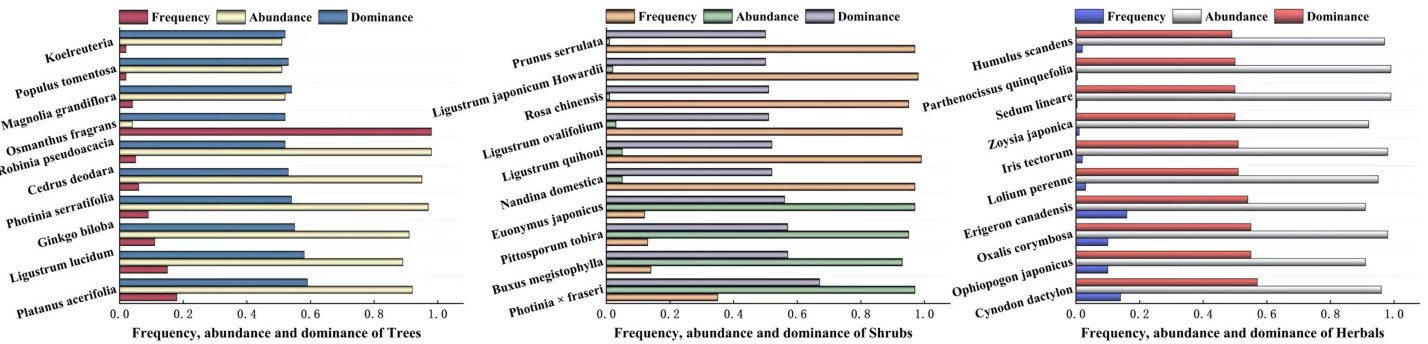

**Figure 4** The top ten plants with dominance of Arbors, Shrubs and Herbals in Zhengzhou. The checklist of the invasive plans of China and Henan determines invasive plants.

The dominance of woody plants and herbaceous plants in Zhengzhou City was calculated using relative frequency, relative plurality, relative significance, or coverage. The results are shown in Fig. 4, which shows the top 10 trees, shrubs, and herbaceous plants in Zhengzhou City. The trees with the highest dominance were *Platanus acerifolia*, *Ligustrum lucidum*, and *Ginkgo biloba*, of which *P. acerifolia* and *G. biloba* are native plants. The three shrubs with the highest dominance were *Photinia × fraseri*, *Buxus megistophylla*, and *Pittosporum tobira*, which are widely cultivated and commonly used for landscaping. Among the top 10 herbaceous plants, *Oxalis corymbosa*, *Erigeron canadensis*, and *Lolium perenne* were invasive plants, whereas *Cynodon dactylon* and *Ophiopogon japonicus* were the most dominant plants.

### Basic characteristics of plant species diversity
#### Different plant groups along the urban–rural gradient

There were certain differences in plant species among the four transects (Fig. 5). Trees were the most unique plants in the western transect at family and species levels. At the genus level, the eastern transect exhibited the highest number of endemic plants, whereas at all three levels, the southern transect showed the lowest number of endemic plants. The number of endemic plants was higher in the eastern and western transects than in the

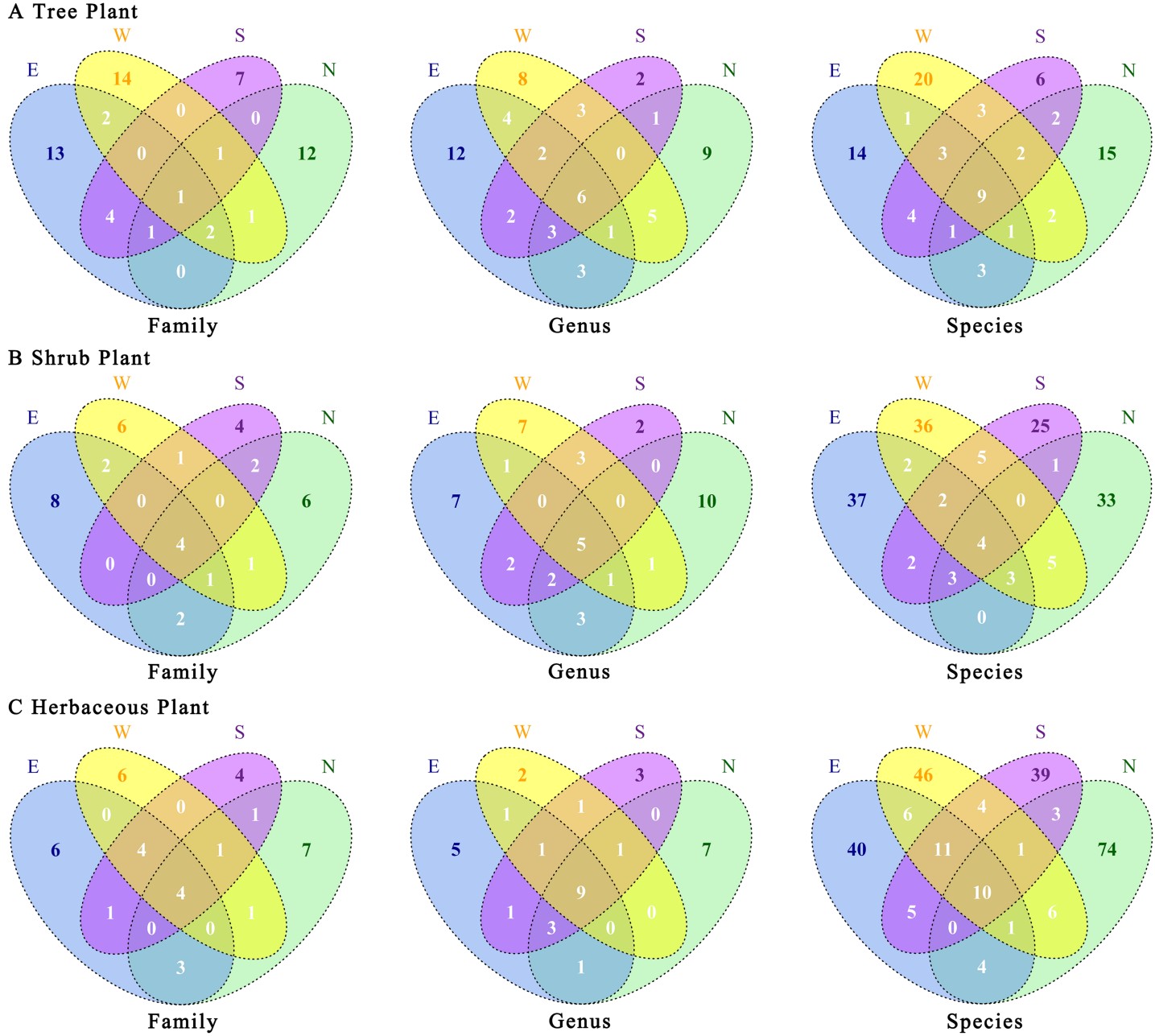

**Figure 5 (A–C) Association of plants' taxonomic categories under the urban–rural gradient in different transects.** E, east transects; W, western transects; S, south transects; N, north transects.

northern and southern transects (Fig. 5A). Shrubs were the most unique plants in the eastern transect at family and species levels. At the genus level, the northern transect exhibited the highest number of endemic plants, whereas the eastern and western transects showed the same number of endemic plants (Fig. 5B). Regarding herbaceous plants, the northern transect showed the highest number of endemic plants at family, genus, and species levels; the southern transect showed the lowest number of endemic plants at family and species levels; and the western transect exhibited the lowest number of endemic plants

at the genus level (Fig. 5C). Overall, the proportions of common plants at genus and species levels increased in the following order: shrubs < trees < herbaceous plants. In contrast, the proportions of common plants at the family level increased in the following order: trees < shrubs = herbaceous plants. Notably, herbaceous plants had a higher proportion of unique species in the four urban–rural gradient sample zones.

Further PCoA analyses of trees, shrubs, and herbaceous plants showed (Fig. 6) that there were no significant differences in total plant species in the four transects. Regarding trees, there were significant differences in plant species between the western transect and other three transects. Regarding shrubs, the plant species in western and northern transects were similar, but they differed significantly compared with those in eastern and southern transects. Regarding herbaceous plants, the plant species in western, southern, and northern transects were similar, with certain differences compared with those in the eastern transect. Overall, the level of species differences decreased in the following order: shrubs > herbaceous plants > trees.

### Plant diversity along different urban–rural gradients

Differences in plant diversity indices were observed in different transects of Zhengzhou City along the urban–rural gradient (Fig. 7). Regarding total species (Fig. 7A), species richness indices were better in the eastern transect than in western, northern, and southern transects. Herbaceous plant diversity indices followed the same pattern as those of total species (Fig. 7D). Regarding trees and shrubs (Figs. 7B and 7C), plant diversity indices were better in the western transect than in eastern, northern, and southern transects. Overall, the plant diversity of eastern and western transects was better than that of northern and southern transects. However, there were no significant differences in the Shannon–Wiener species diversity index, the Simpson diversity index, and Pielou's species evenness index of the plants.

## Characterization of plant diversity distribution patterns along the urban–rural gradient

Varying types of plants displayed distinct distribution patterns along W–E and S–N axes (Fig. 8A). In terms of the species richness index, all plants and spontaneous plants in the eastern transect (Figs. 8A and 8O), trees and spontaneous plants in the western transect (Figs. 8C and 8O), and shrubs and cultivated plants in the southern transect (Figs. 8F and 8N) exhibited an unimodal spatial pattern. All plants, herbaceous plants, and exotic plants in the western transect (Figs. 8A, 8G, and 8K); trees in the southern transect (Fig. 8D); trees, shrubs, native plants, and cultivated plants in the northern transect (Figs. 8D, 8F, 8J, and 8N); and shrubs in the eastern transect (Fig. 8E) exhibited a quadratic polynomial pattern, with a decreasing pattern from the city center to the suburbs. Meanwhile, herbaceous, exotic, and spontaneous plants exhibited a quadratic polynomial pattern in the northern transect, with an increasing pattern from the city center to the suburbs (Figs. 8H, 8L, and 8P). All plants, herbaceous plants, exotic plants, and spontaneous plants in the southern transect (Figs. 8B, 8H, 8L, and 8P) as well as herbaceous plants, native plants, exotic plants, cultivated plants, and spontaneous plants in the eastern transect (Figs. 8G, 8I,

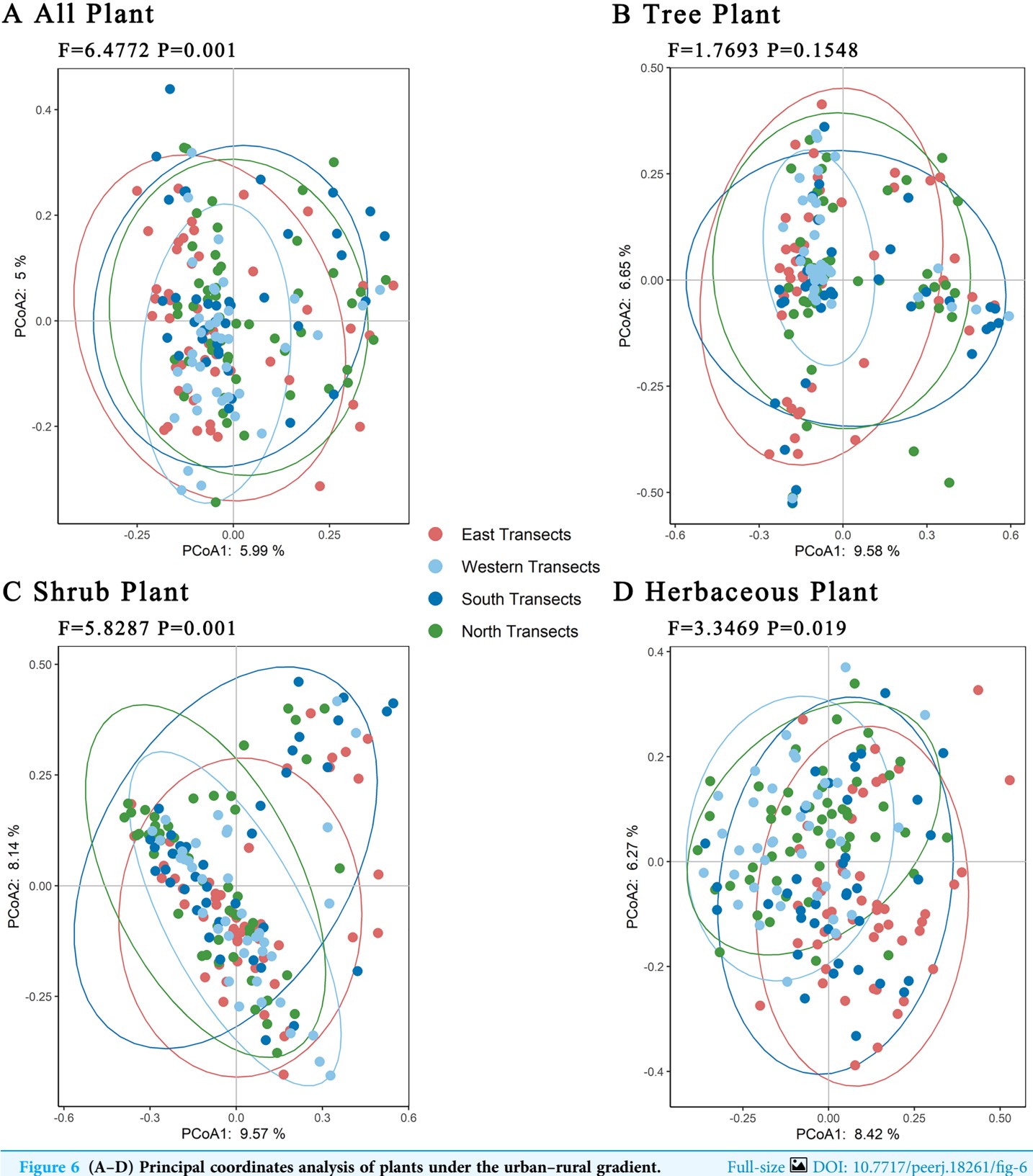

**Figure 6** (A–D) Principal coordinates analysis of plants under the urban–rural gradient.

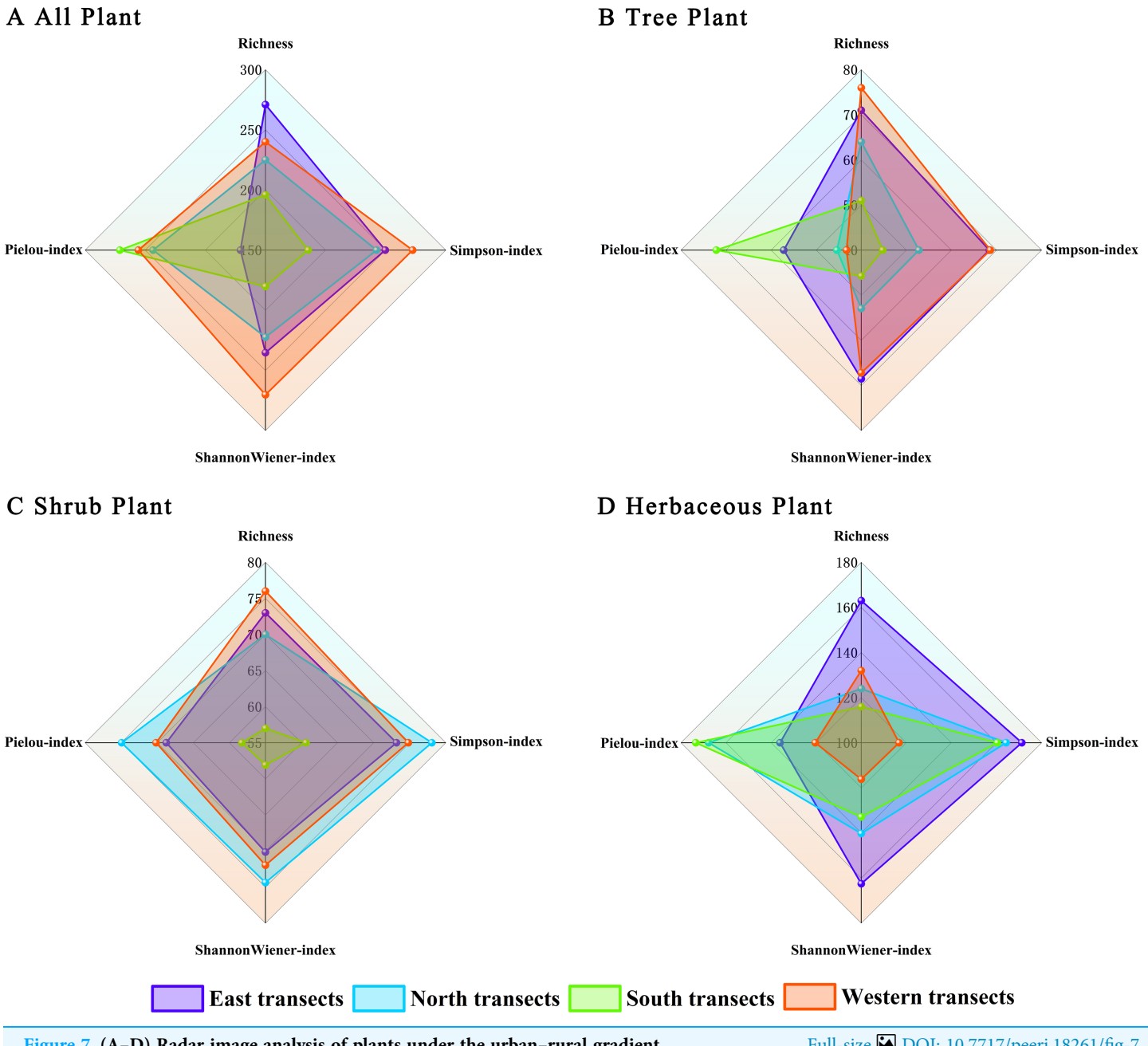

**Figure 7 (A–D) Radar image analysis of plants under the urban–rural gradient.**

8K, 8M, and 8O) exhibited a cubic polynomial pattern, showing a spatial pattern of first decreasing, then increasing, and finally decreasing from the city center to the rural area. Trees in the eastern transect (Fig. 8C) as well as shrubs, native plants, and cultivated plants in the western transect (Figs. 8E, 8I, and 8M) followed a cubic polynomial pattern. However, they showed a spatial pattern of first increasing, then decreasing, and finally increasing from the city center to the suburbs. No specific quantitative pattern was noted for all plants in the northern transect or native plants in the southern transect (Figs. 8B and 8J).

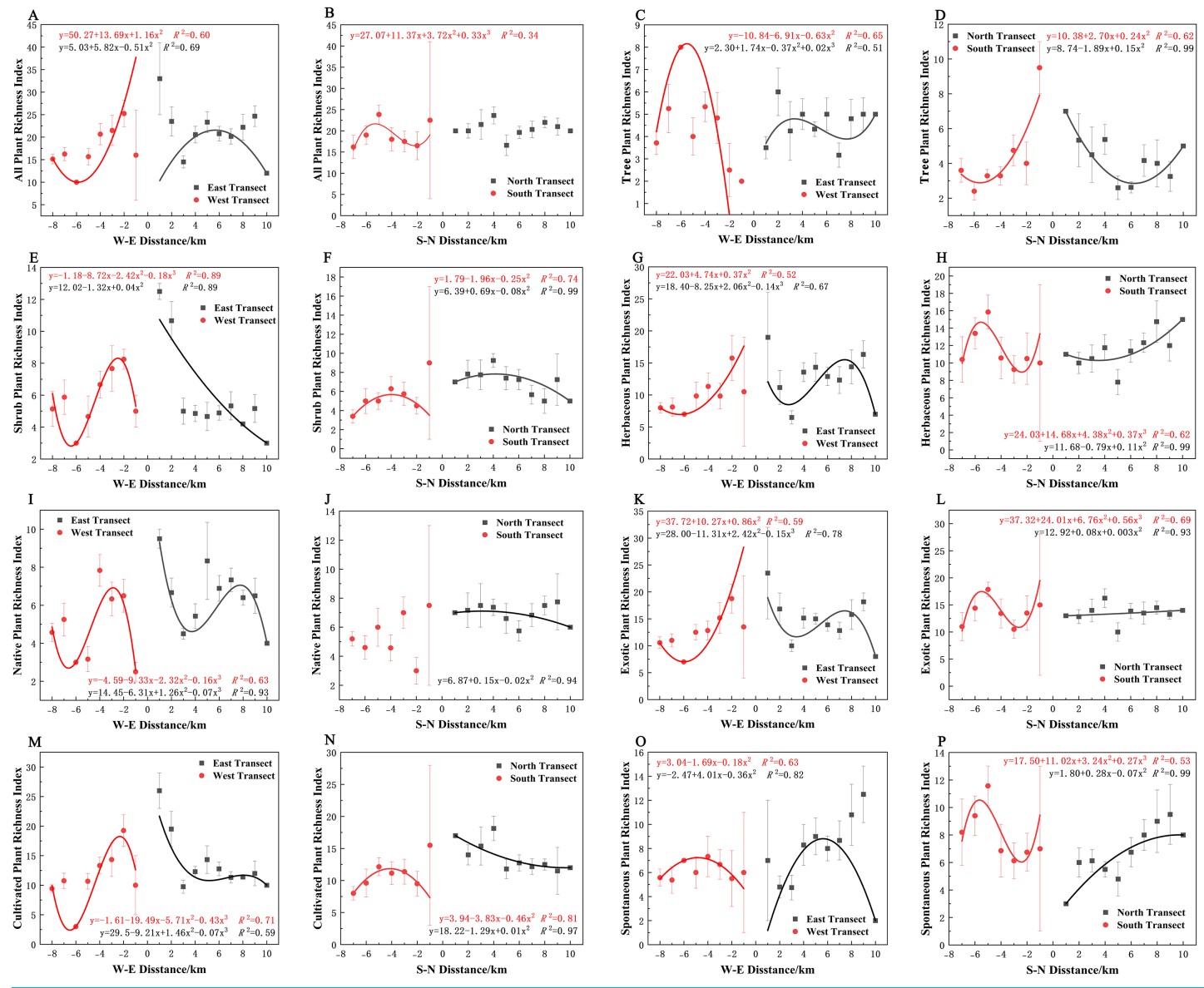

**Figure 8 Spatial pattern of plant species richness along the west-east (W–E) and south-north (S–N) axes.** (A, B) All plants, (C, D) tree plant, (E, F) shrub plant, (G, H) herbaceous plant, (I, J) native plant, (K, L) exotic plant, (M, N) cultivated plant, (O, P) spontaneous plant.

In terms of the Shannon–Wiener diversity index, all plants, shrubs, and native and cultivated plants in the eastern and western transects (Figs. 9A, 9E, 9I, and 9M) and trees in the southern and northern transects (Fig. 9D) exhibited a quadratic polynomial pattern, with an overall decreasing pattern from the city center to the suburbs. All plants, shrubs, native plants, and cultivated plants in the southern transect (Figs. 9B, 9F, 9J, 9N) as well as trees and spontaneous plants in the western transect (Figs. 9C and 9O) exhibited an unimodal pattern. Herbaceous plants, native plants, exotic plants, and cultivated plants exhibited a quadratic polynomial pattern in the northern transect, with an increasing pattern from the city center to the suburbs. All plants and shrubs in the northern transect

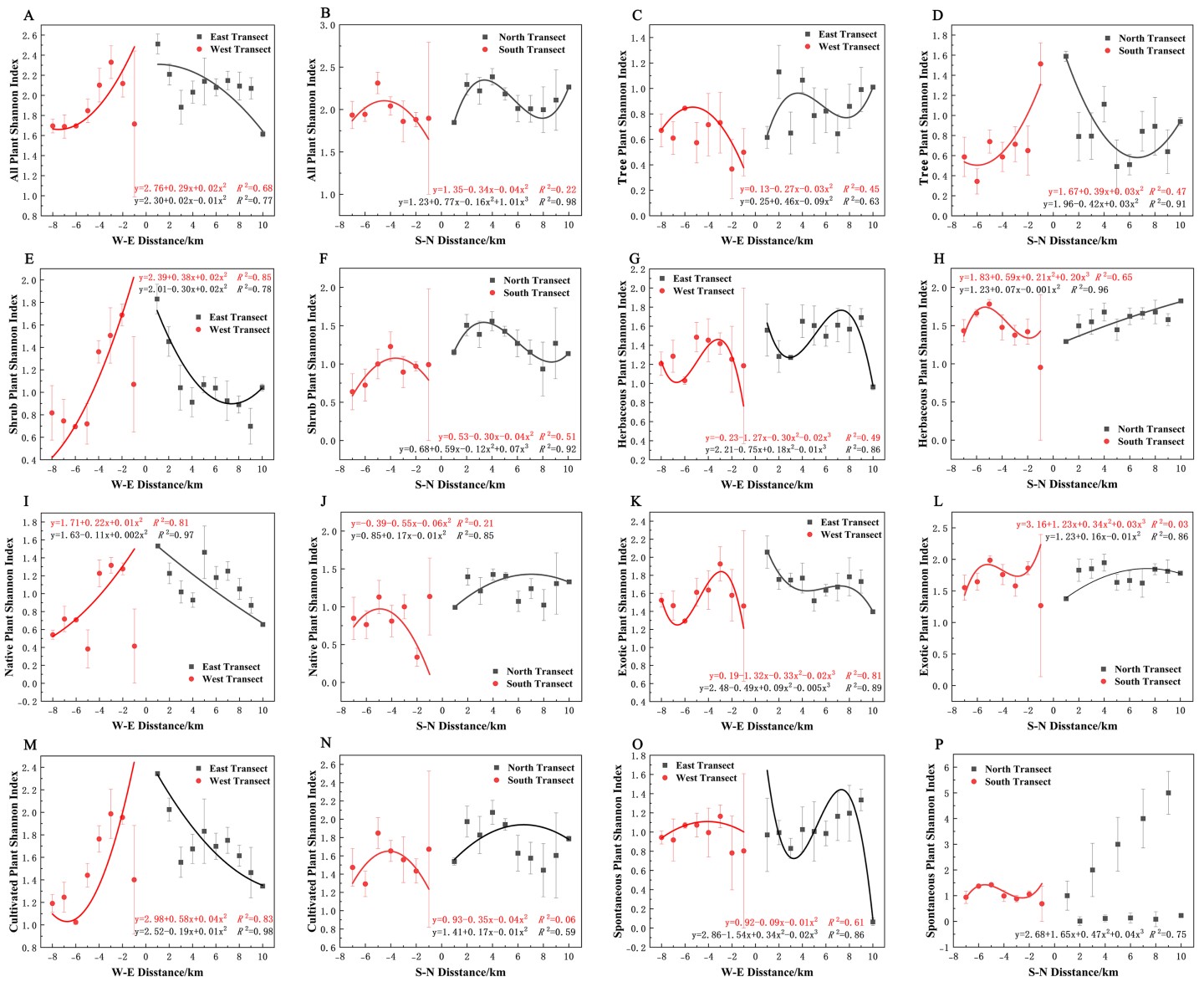

**Figure 9** (A–P) Spatial pattern of Shannon–Wiener diversity of plant species, with letters corresponding to the legend in **Fig. 8**.

(Figs. 9B and 9F), trees in the eastern transect (Fig. 9C), and herbaceous and exotic plants in the western transect (Figs. 9G and 9K) exhibited a cubic polynomial pattern showing a spatial pattern of first increasing, then decreasing, and finally increasing from the city center to the suburbs. Furthermore, herbaceous plants, exotic plants, and spontaneous plants exhibited a cubic polynomial pattern in eastern and southern transects (Figs. 9G, 9H, 9K, 9L, 9O, and 9P); however, the overarching trend is a decline from the urban center to the outskirts of the city, followed by an increase and subsequent decline. However, no clear pattern of spontaneous plants was noted in the northern transect (Fig. 9P).

In terms of the Simpson diversity index, all species, shrubs, native plants, exotic plants, and cultivated plants in the eastern transect (Figs. 10A, 10E, 10I, 10K and 10M); all plants,

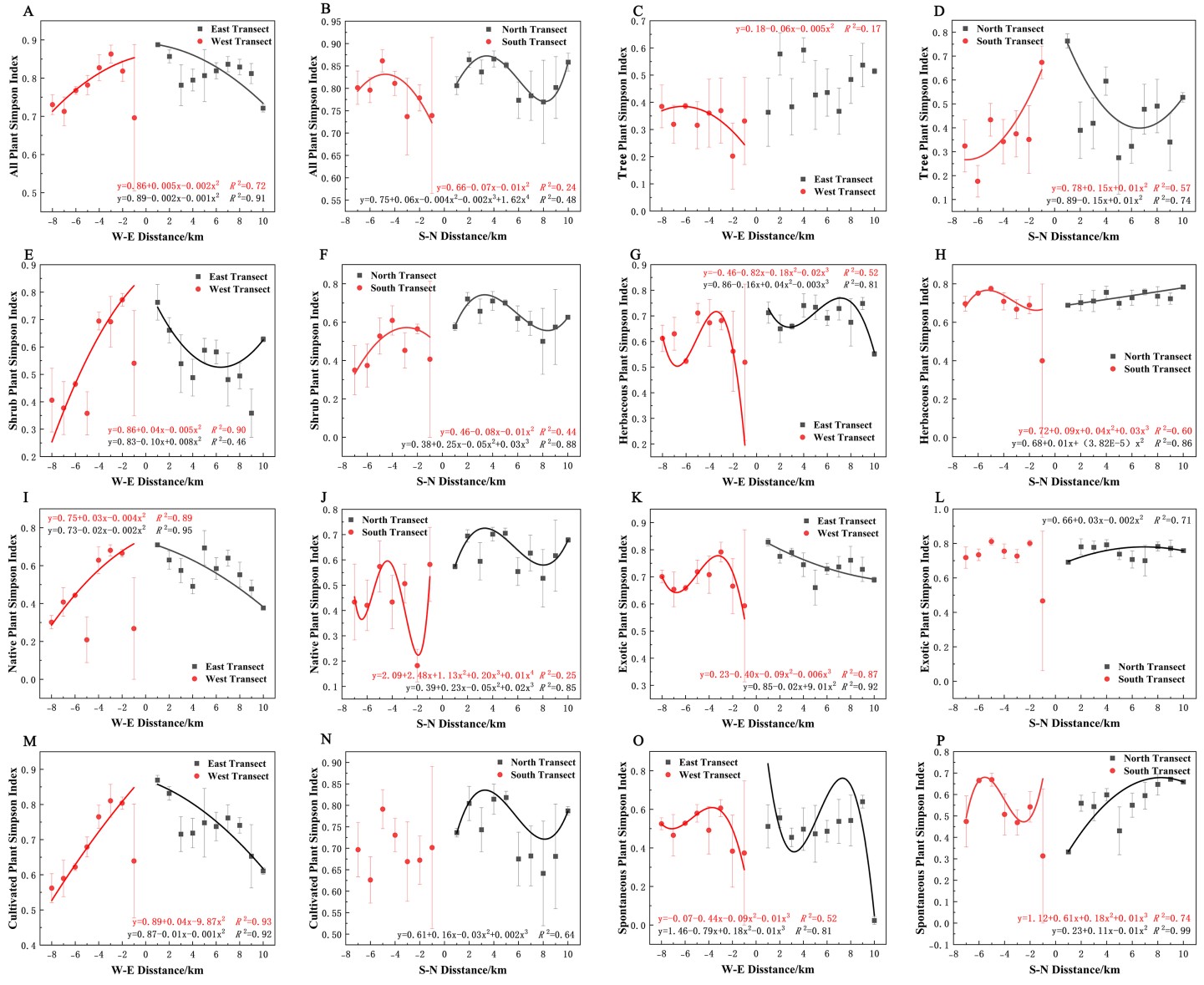

**Figure 10 (A–P) Spatial pattern of Simpson's diversity of plant species, with letters corresponding to the legend in Fig. 8.**

trees, shrubs, native plants, and cultivated plants in the western transect (Figs. 10A, 10C, 10E, 10I and 10M); and trees in the S–N transect (Fig. 10D) exhibited a quadratic polynomial pattern. The overall pattern was linearly decreasing from the city center to the outskirts. All plants and shrubs showed a single-peaked spatial pattern in the southern transect and trees in the western transect (Figs. 10B, 10F, and 10C). Herbaceous, exotic, and spontaneous plants all exhibited a quadratic polynomial pattern in the northern transect (Figs. 10H, 10L, and 10P), with a linear increasing pattern from the city center to the outskirts. All plants, shrubs, native plants, and cultivated plants in the northern transect (Figs. 10B, 10F, 10J, and 10N). Herbaceous plants, exotic plants, and spontaneous plants in the western transect (Figs. 10G, 10K, and 10O) exhibited a cubic polynomial

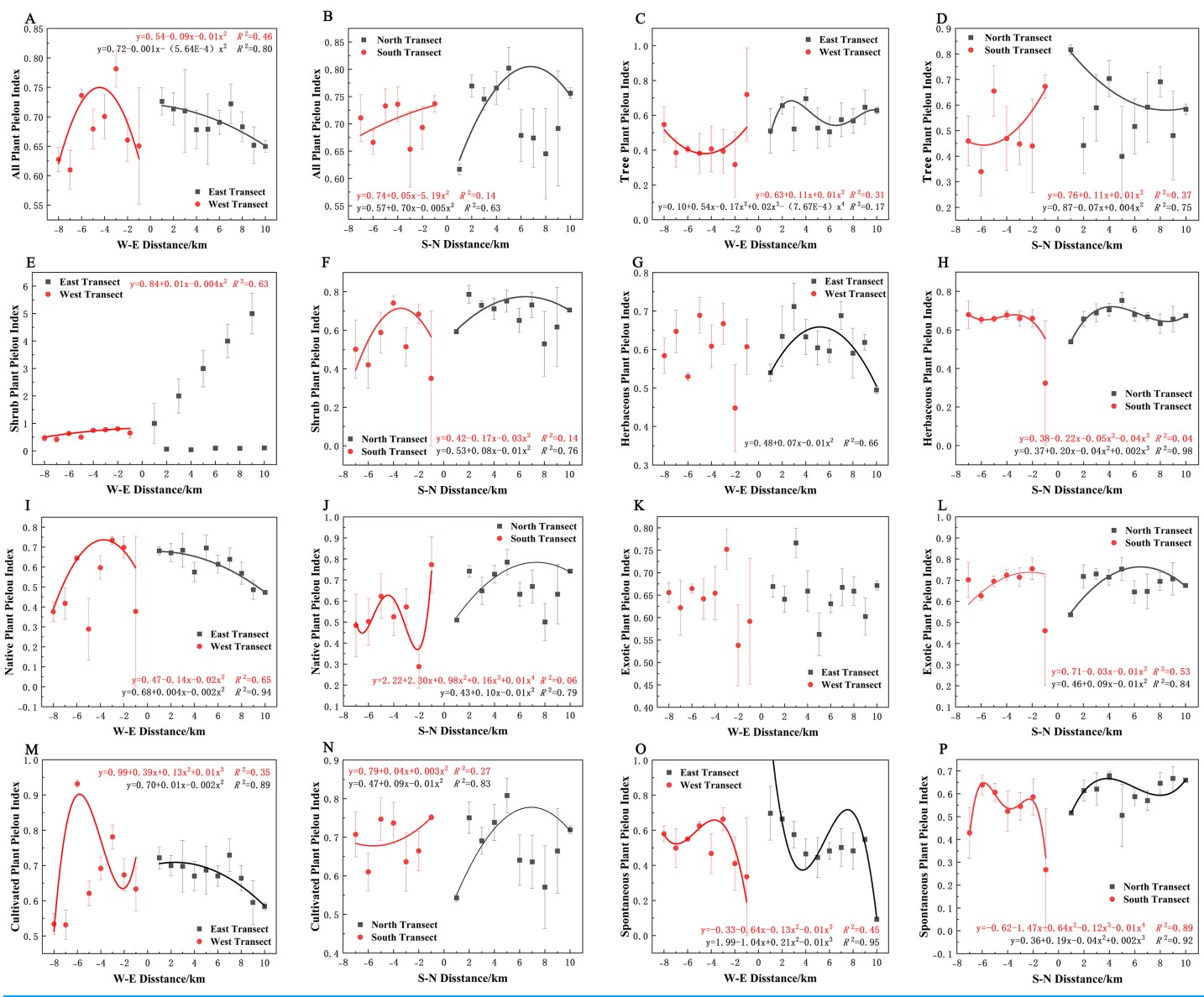

**Figure 11** (A–P) Spatial pattern of plant species evenness, with letters corresponding to the legend in **Fig. 8**.

pattern, showing a spatial pattern of first increasing, then decreasing, and finally increasing from the city center to the outskirts. Native and spontaneous plants also exhibited a cubic polynomial pattern in the eastern and southern transects (Figs. 10G, 10H, 10O, and 10P), showing a spatial pattern of first decreasing, then increasing, and finally decreasing from the city center to the outskirts. Native plants in the southern transect exhibited a fourth-order polynomial pattern (Fig. 10J). In contrast, trees in the eastern transect and exotic plants and cultivated plants in the southern transect (Figs. 10C, 10L, and 10N) did not have specific numerical patterns.

In terms of Pielou's evenness index, all plants, native plants, and cultivated plants in the eastern transect (Figs. 11A, 11I, and 11M); all plants, trees, exotic plants, and cultivated

plants in the southern transect (Figs. 11B, 11D, 11L, and 11N); trees and shrubs in the western transect (Figs. 11C and 11E); and trees in the northern transect (Fig. 11D) exhibited a quadratic polynomial pattern, with a linear decreasing pattern from the city center to the suburbs. All plants and native plants in the western transect (Figs. 11A and 11I); all plants, shrubs, native plants, exotic plants, and cultivated plants in the northern transect (Figs. 11B, 11F, 11J, 11L and 11N); shrubs in the southern transect (Fig. 11F); and herbaceous plants in the eastern transect (Fig. 11G) exhibited a unimodal spatial pattern. Herbaceous and spontaneous plants in the northern transect (Figs. 11H and 11P), herbaceous plants in the southern transect (Fig. 11H), and spontaneous plants in the western transect (Fig. 11O) exhibited a cubic polynomial pattern, showing an overall spatial pattern of first increasing, then decreasing, and finally increasing from the city center to the outskirts. Cultivated plants in the western transect (Fig. 11M) and spontaneous plants in the eastern transect (Fig. 11O) exhibited a cubic polynomial pattern, showing a spatial pattern of first decreasing, then increasing, and finally decreasing from the city center to the suburbs. Trees in the eastern transect (Fig. 11C) and spontaneous and native plants in the southern transect exhibited a quartic polynomial spatial pattern (Figs. 11P and 11J). However, shrubs in the eastern transect (Fig. 11E), herbaceous plants in the western transect (Fig. 11G), and exotic plants in eastern and western transects (Fig. 11K) did not exhibit clear patterns.

## Impact of factors on plant diversity in the context of urbanization

Results of the correlation test between the richness of different types of plants and environmental factors (Fig. 12) showed that the species richness of all plants, herbaceous plants, and native plants were significantly positively correlated with the landscape shape index (LSI) and house prices (HP) ($p < 0.05$) and significantly negatively correlated with $Sealed_{500}$. The abundance of shrubs and cultivated plants was significantly and positively correlated with GDP. HP was positively correlated with the effect of exotic plant richness and significantly negatively correlated with $Sealed_{500}$. Spontaneous plant richness was significantly and positively correlated with HP and significantly negatively correlated with night light index (NL). Overall, there was some variation in the drivers of plant richness across different types of plants.

There was some variation in the drivers of richness at the patch scale across different types of plants. The analyses (Table 3) showed that the optimal model for total species had a total explanatory rate of 21% for richness variation. GDP, HP, and spatial distribution of population (SDP) were significantly and positively correlated with total species richness ($p < 0.05$), whereas $Sealed_{500}$ and NL were negatively correlated with total species richness. The best-fit model for tree richness explained 5% of the total variation in richness, with BD being significantly and positively correlated with tree richness ($p < 0.05$) and GDP and $Sealed_{500}$ being negatively correlated with tree richness. The best model for shrub species richness explained 16% of the total variation, with BD and GDP being significantly and keenly correlated with shrub richness ($p < 0.05$). In contrast, NL was negatively correlated with shrub richness. The optimal model for herbaceous plants had a total explanatory rate of 17% for richness variation, with GDP and HP being significantly and positively

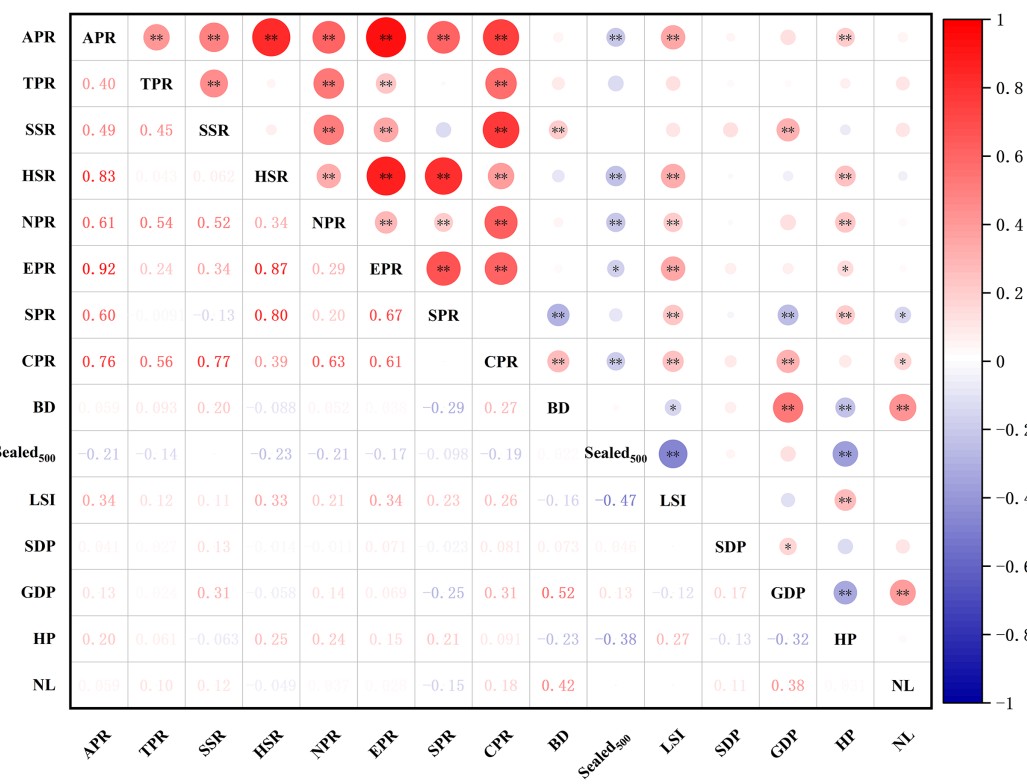

**Figure 12 Spearman correlations analysis of different groups of species richness and environment variables.** Abbreviations please see Table 2. In the picture: (−) means negative correlation; asterisk (* and **) indicate the significance at $p = 0.05$ and $p = 0.01$.

**Table 3 Prediction results of species richness of different categories at patch scale using generalized linear models.**

| Variables | Equation | Parameters |
|---|---|---|
| All plant | y = 2.99 + 0.07GDP + 0.09HP + 0.04SDP − 0.02Sealed$_{500}$ + 0 .01BD + 0.01LSI − 0.02NL | $R^2 = 0.21$ |
| Tree plant | y = 1.45 + 0.08BD + 0.02HP + 0.04SDP − 0.01Sealed$_{500}$ + 0.01LSI − 0.01GDP | $R^2 = 0.05$ |
| Shrub plant | y = 1.81 + 0.09GDP + 0.05HP + 0.07SDP + 0.01Sealed$_{500}$ + 0.1BD − 0.01NL | $R^2 = 0.16$ |
| Herbaceous plant | y = 2.43 + 0.07GDP + 0.08HP − 0.06Sealed$_{500}$ − 0.02BD + 0.01LSI − 0.04NL | $R^2 = 0.17$ |
| Native plant | y = 1.81 + 0.07GDP + 0.1HP + 0.01LSI | $R^2 = 0.09$ |
| Exotic plant | y = 2.62 + 0.07GDP + 0.08HP + 0.05SDP − 0.02Sealed$_{500}$ − 0.01NL | $R^2 = 0.15$ |
| Cultivated plant | y = 2.53 + 0.07GDP + 0.09HP + 0.05SDP − 0.02Sealed$_{500}$ + 0.12BD + 0.02LSI − 0.01NL | $R^2 = 0.29$ |
| Spontaneous plant | y = 1.96 + 0.08HP − 0.01Sealed$_{500}$ − 0.14BD + 0.01LSI − 0.03NL | $R^2 = 0.22$ |

correlated with herbaceous plant richness ($p < 0.05$) and Sealed$_{500}$, BD, and NL being negatively correlated with herbaceous plant richness.

The best-fit model for the pattern of native plant richness explained 9% of the total variation in richness, with GDP and HP being significantly and positively correlated with native plant richness ($p < 0.05$). In contrast, the best-fit model for exotic plant richness

explained 15% of the total variation in richness, with GDP, HP, and SDP being significantly and positively correlated with exotic plant richness ($p < 0.05$) and Sealed$_{500}$ and NL being negatively correlated with exotic plant richness.

The best-fit model for the richness of cultivated plant species exhibited a total explanatory rate of 29% for the variation in richness. BD, GDP, HP, and SDP were significantly positively correlated with cultivated plant richness ($p < 0.05$), whereas Sealed$_{500}$ and NL were negatively correlated with cultivated plant richness. The optimal model for spontaneous plant species richness explained 22% of the total variation. HP displayed a significant positive correlation ($p < 0.05$) with spontaneous plant species richness, and BD exhibited a significant positive correlation ($p < 0.05$) with spontaneous plant species richness.

The relative contribution rates of various environmental factors are shown in Fig. 13. Regarding total species richness, the model effect explained 21% of the variation, with HP independently explaining 7% of the variation, accounting for 33.3% of the total explanatory rate. GDP and SDP independently explained 6% and 3% of the variation, respectively, and the explanatory rates of other variables were <2%. The model effect explained 5% of the variation in tree richness, with BD explaining 3% of the total variation, accounting for 60% of the total explanatory rate of the model effect. The explanatory rates of other variables were all <1%. Regarding shrub species richness, the model effect explained 16% of the variation, with BD explaining 5% of the total variation, accounting for 31.3% of the total explanatory rate. GDP and SDP independently explained 4% and 3% of the variation, respectively, and the explanatory rates of other variables were all <2%. The model effect explained 17% of the variation in herbaceous plant richness, with HP explaining 5% of the total variation, accounting for 29.4% of the total explanatory rate of the model effect. GDP and Sealed$_{500}$ independently explained 4% of the variation, whereas the explanatory rates of other variables were all <2%.

The model effect explained a total of 9% of the variance in the abundance of native plants, of which HP explained 5% of the total variance, accounting for 55.6% of the total explanatory rate of the model effect. GDP independently explained 3% and the explanatory rates of other variables all <1%. The variation in the richness of exotic plant species could be explained by model effects by a total of 15%, with HP accounting for 5% of the total variation, accounting for 33.3% of the total explanatory rate. GDP can independently explained 4%, SDP 3%, and the explanatory rates of other variables all <2%.

The model effect explained a total of 29% of the variance in cultivated plant richness, with BD explaining 10% of the total variance, accounting for 34.5% of the total explanatory rate of the model effect, HP 7%, GDP 6%, SDP 4%, and the remaining variables <1% of the total variance. The model effect explained a total of 22% of the variance in spontaneous plant richness, with BD explaining 11% of the total variance, accounting for 50% of the total explanatory rate, HP 7%, and the remaining variables <2%. Based on these results, four variables, GDP, HP, SDP, and BD, contributed the most to the best model explanatory rate among all plant types.

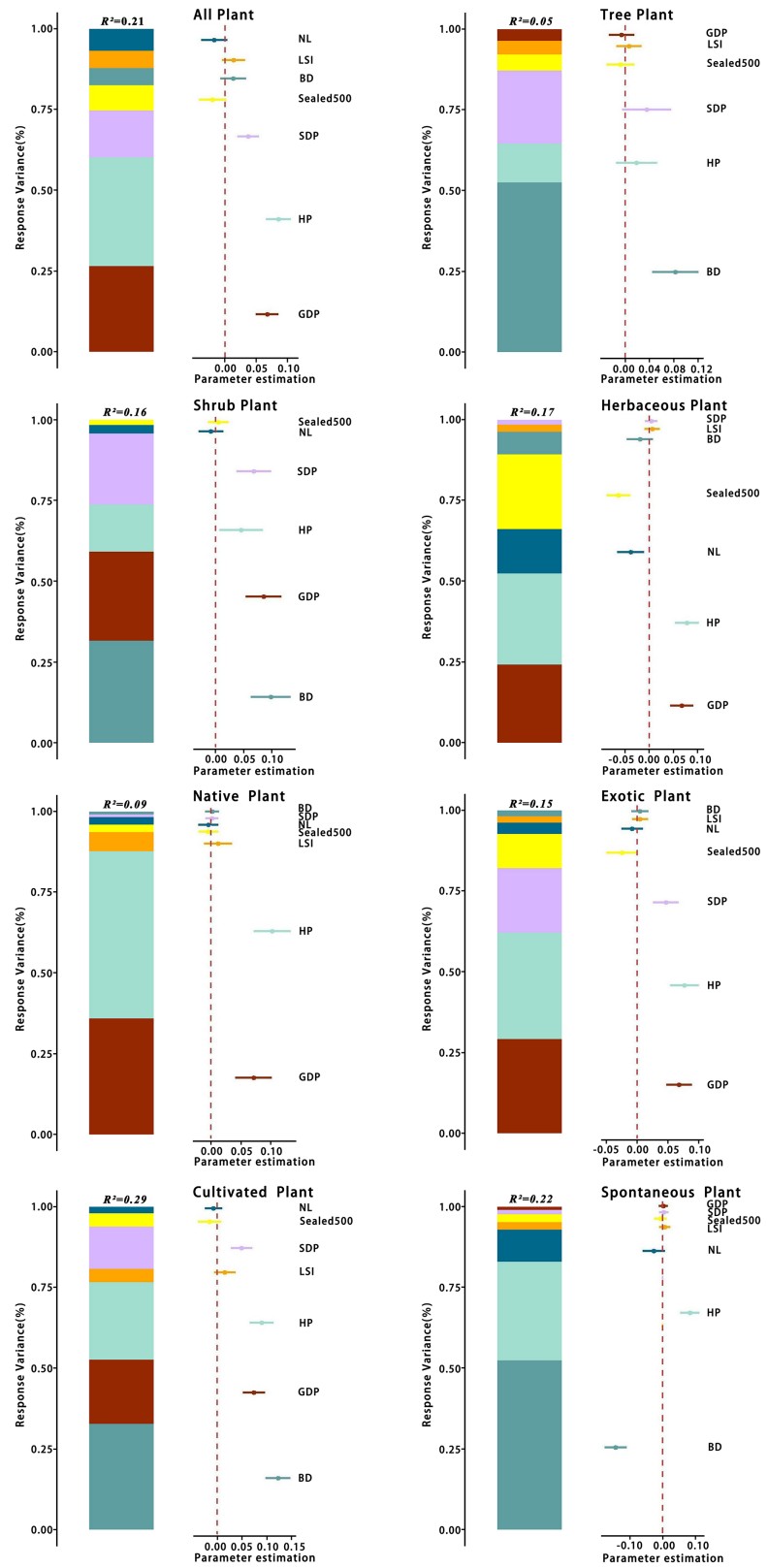

**Figure 13 Variance partition of species richness for different categories at the patch scale.** For abbreviations, please see Table 2.         

## DISCUSSION

### Characteristics of plant diversity in Zhengzhou City

In this survey, the richness of herbaceous plants (366 species) in the built-up area of Zhengzhou City was considerably higher than that of trees (106 species) and shrubs (124 species), and it is necessary to focus on the importance of herbaceous plants in the diversity of urban green spaces (Norton et al., 2019). There were 416 cultivated plants in Zhengzhou, accounting for 69.80% of the total species, indicating that cultivated plants are an important component of the urban flora (Zhang et al., 2023). Although urban greening strongly advocates the use of native plants, the number of cultivated plants introduced from abroad (74.32%) was considerably higher than that of locally cultivated plants (25.90%). Among foreign plants, garden plants accounted for a substantial proportion, with a total of 134 species. The introduction of exotic species may increase biodiversity for a certain period, but their extensive use and gradual domestication may exacerbate the homogenization of urban plant communities (Li et al., 2019). Therefore, the application of functional traits of native plants should be emphasized in the greening and design process of the built-up area of Zhengzhou City (Yang et al., 2017). In contrast, specific functional traits can be targeted to provide ecosystem services required for urban green spaces (Fan, Liang & Li, 2021).

In the urban artificial plant community environment, wild or spontaneous plants are more likely to occur in the herbaceous layer than in the woody layer, making the herbaceous layer richer in plant species (Levine & HilleRisLambers, 2009). In addition to garden cultivars, 180 spontaneous plant species were surveyed in the herbaceous layer, accounting for ~50% of the herbaceous plants. This demonstrated that spontaneous plants had become an organic part of the low-maintenance plant landscapes in the construction of urban green spaces. Herbaceous plants, the lower layer of the community, are characterized by better soil conditions and nutrient substrates and are therefore most likely to undergo a near-natural succession of plant communities, forming a low-maintenance biodiversity (Yang et al., 2023). Wild plants cannot be easily pulled out, have high biodiversity, and play an important role in the ecosystem. Park management should pay attention to the ecological benefits of wild weeds (Powell & Lenton, 2013).

### Patterns of plant diversity distribution along the urban–rural gradient

These findings quantitatively analyzed the pattern of change in plant species diversity across the urban–rural continuum in Zhengzhou City. The distribution patterns of different plant taxa along the urban–rural gradient showed anisotropic heterogeneity, with the coexistence of multiple spatial patterns. This finding enriched the theoretical system of plant diversity response to the urban–rural gradient. The urban–rural gradient showed the highest total species richness in each category between 2 and 10 km from the city center in the four sample transects, consistent with the findings of Tian, Song & Da (2015) that the central urban area had the highest level of plant richness. This may be attributed to the moderate level of urbanization in the region, which allows the escape of some plants carried or cultivated by humans, increasing species richness to some extent. The richness

of trees, shrubs, and herbaceous plants was also higher in the central urban area, as reported in the Beijing study (*Wang, MacGregor-Fors & López-Pujol, 2012*). This was mainly because most of the green spaces in the city are artificially vegetated, and woody plants with long life histories and perennial herbaceous plants are often selected due to human preferences and plant configurations created by landscape architects (*Clarke, Jenerette & Davila, 2013*). In contrast, annual herbaceous plants in the city center predominantly include weeds and exotic flowers with low species richness because of intensive green space management activities (*Zhang, Zhou & Ma, 2020*). The higher species richness of exotic plants across the sample zones may be influenced by the high number of exotic species introduced into the city center, and other studies have confirmed that human introduction of exotic plants increases species richness (*Dylewski et al., 2023*; *Egawa & Koyama, 2023*).

The varying patterns of plant diversity along the urban–rural gradient can be attributed to the following: (1) landscapes in the city exhibit a great extent of discrepancy, providing a rich choice of habitats for various plant species (*Hammill et al., 2018*); (2) urban areas have various land-use types, and the variety of land-use forms and spatial configurations contribute to the enhancement of plant diversity to some extent (*Felipe-Lucia et al., 2020*); (3) throughout the urban–rural transition zone, the socioeconomic conditions, land-use management level, cultural diversity, and extent of anthropogenic disturbance in cities also influence plant diversity distribution patterns (*Cui et al., 2023*; *Peng et al., 2019*).

## Influence of urban environmental variables on the diversity of different types of plants

After the reform and opening up, Zhengzhou City has experienced rapid urbanization, with the expansion of urban land types and landscape fragmentation greatly impacting biodiversity, showing an increase in plant species richness and changes in structure and function (*Hou et al., 2023*; *Nock et al., 2013*; *Zhang, Song & Da, 2020*). The homogenization of plant communities in Zhengzhou City has severely affected biodiversity, and this study focused on the impact of urbanization factors on biodiversity (*Bergey & Whipkey, 2020*; *Zeeman et al., 2017*).

In this study, plant richness in each category was significantly and positively correlated with LSI. An increase in species richness with increasing complexity of patch shapes was also observed in urban biodiversity studies in Henan, Chongqing, and Gwangju, South Korea (*Gao et al., 2021*; *Kim & Pauleit, 2005*; *Qi et al., 2024*; *Wang et al., 2020a*). Although $Sealed_{500}$ was negatively correlated with species richness, our findings were consistent with those from Haifa, Israel (*Malkinson, Kopel & Wittenberg, 2018*) and Germany (*Albrecht & Haider, 2013*). This was because increased imperviousness makes connectivity between patches weaker, further making species dispersal more challenging, thereby reducing species richness. Second, increased imperviousness represented increased disturbance, which could also decrease species richness. NL has been widely used to reflect the intensity of urbanization in recent years (*Cui et al., 2019*), which is negatively correlated with species richness, probably because a higher NL represents greater human disturbance, leading to a decrease in species richness.

There were also differences in the explanatory rates of different plant types according to each influencing factor, with total species richness being influenced by GDP, HP, SDP, and distance of patches from the urban edge (BD). Tree and shrub richness was influenced by BD, whereas shrub richness was influenced by GDP, HP, and SDP (*Gao et al., 2023*). Herbaceous plant richness was influenced by GDP, HP, imperviousness within 500 m from the patch, and NL. Native plant richness was mainly influenced by GDP and HP (*Wang et al., 2023*). In contrast, exotic and cultivated plant richness were influenced by the factors affecting total species richness and shrub species richness, respectively. Spontaneous plant richness was influenced by HP, BD, and NL. Overall, GDP, HP, SDP, and BD significantly affected the pattern of plant richness in each category.

### Limitations and outlook

During the survey and data collection process, some trees with small diameters at breast height and low height were recorded as shrubs, and some shrubs were recorded as trees, which affected the number of tree and shrub species to some extent. In addition, the different planting methods, such as *Photinia serratifolia* and *B. megistophylla* planted in patches, exhibited a higher count per square unit, whereas the lower density of *Rosa chinensis* possessed a reduced density per unit area, which may affect the results of the dominant species (*Wang et al., 2020b*). If similar studies were carried out in the future, more scientific research standards should be developed to reduce manual errors and ensure the accuracy of the results. This study only compared and analyzed the plant diversity in different transects of the built-up area of Zhengzhou City. Future research should focus on exploring the potential mechanisms between plant diversity in different regions and the degree of urbanization in different periods, to guide the improvement of plant diversity across various locations (*Fan et al., 2019*; *Li et al., 2019*).

## CONCLUSIONS

Our study focused on the plant diversity status in Zhengzhou City, differences in plant diversity across different transects, changes in distribution patterns, and factors influencing plant diversity. Understanding these concepts may help understand the current status of plant ecosystems in Zhengzhou City and provide references for urban green space planning.

Different types of plants in the built-up area of Zhengzhou City showed obvious differences in diversity. As the main component of the urban green landscape, trees had the richest functional diversity, shrubs had the greatest species differences and enriched the ornamental properties of plant species in each sampling plot, and various spontaneous plants appeared in the herbaceous layer, thus improving species diversity. Cultivated plants are an important component of the urban flora, and future studies should focus on conferring increased protection to spontaneous plants. In the urban green space, native plants accounted for 24.5% and exotic plants accounted for 75.5% of the total variation, supplementing the missing functional characteristics in native plants. Along the urban–rural gradient, differences were noted in the distribution pattern of diversity of different plant groups. Overall, the richness of plant diversity was the highest in the central

urban area, whereas the richness of spontaneous plant species was the highest in the suburbs. This finding was related to the lower level of maintenance and management in the suburbs. The richness of plant species in different plant types is influenced by urbanization factors, and different measures are required to effectively improve plant diversity. These research findings can serve as a basis for the planning of urban green spaces and conservation of plant diversity in Zhengzhou.

### Funding
This work was supported by the Research Project of Henan Provincial Federation of Social Sciences, grant number (SKL-2023-2598). The funders had no role in study design, data collection and analysis, decision to publish, or preparation of the manuscript.

### Grant Disclosures
The following grant information was disclosed by the authors:
Research Project of Henan Provincial Federation of Social Sciences: SKL-2023-2598.

### Competing Interests
Chong Du is employed by SuperMap Software Co., Ltd.

### Author Contributions
- Lingling Zhang conceived and designed the experiments, performed the experiments, analyzed the data, prepared figures and/or tables, authored or reviewed drafts of the article, and approved the final draft.
- Chong Du analyzed the data, prepared figures and/or tables, and approved the final draft.
- Wenhan Li conceived and designed the experiments, authored or reviewed drafts of the article, and approved the final draft.
- Yongjiang Liu performed the experiments, prepared figures and/or tables, and approved the final draft.
- Ge Zhang performed the experiments, prepared figures and/or tables, and approved the final draft.
- Shanshan Xie conceived and designed the experiments, authored or reviewed drafts of the article, and approved the final draft.
- Yiping Liu conceived and designed the experiments, analyzed the data, authored or reviewed drafts of the article, and approved the final draft.
- Dezheng Kong conceived and designed the experiments, authored or reviewed drafts of the article, and approved the final draft.

### Data Availability
The raw data are available in the Supplemental File.

## Supplemental Information

Supplemental information for this article can be found online at http://dx.doi.org/10.7717/peerj.18261#supplemental-information.

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
