# Peer review of "Spatial patterns and driving factors of plant diversity along the urban–rural gradient in the context of urbanization in Zhengzhou, China"

_PeerJ, doi:10.7717/peerj.18261_

## Round 0.1 · original submission · Major Revisions

One reviewer requested minor changes, but the other reviewer rejected the paper. In all, the paper requires major changes. Reviewer 2 in particular points out a number of major issues with the paper, of which the major ones include research design, data presentation and analysis.

Please revise the paper, and provide a detailed rebuttal of each aspect raised by both reviewers.

·

Basic reporting

There are literature references and sufficient field background/context provided.
It has a professional article structure, figures, tables.
Self-contained with relevant results to hypotheses

Experimental design

Research question well defined, relevant & meaningful. It is stated how research fills an identified knowledge gap.
The methods are described with sufficient detail & information to replicate.

Validity of the findings

All underlying data have been provided; they are robust, statistically sound, & controlled.

Additional comments

no comment

Reviewer 2 ·

Basic reporting

This study aimed to study the distribution, composition and diversity of plants along urban gradients. The question is not new and the introduction fails to clearly clarify how this study is original, how it fits into the context of what has already been done, there are no hypotheses formulated. The English should really be improved, many sentences have no verbs (e.g. L112, L131, L195…), formulations and words are often not appropriate (e.g replace the word arbor with tree, L163- 165 incomprehensible sentence). Significant field work was carried out, 178 sites were sampled but the design and analysis are insufficiently described.The review was not continued for the results and discussion sections because it was too difficult to read.

Experimental design

Sampling method is imprecise and some sources used are missing.
e.g. 108-109 “According to the statistics, as of 2022, the green space rate in Zhengzhou City is 36.81%” what is the source?
e.g. 113-115 Is it stratified and by what? Or is it random as stated in the same sentence? What are the different types of habitats analyzed? L 118 Why did you not conserve area without vegetation?
e.g. Table 1 The description of the data collected is too brief, some information is missing, what is “area”, the percentage of plant cover? A surface? For herbaceous plants, height is measured on one individual or more, is it a mean? And when in your paper this measure of height is used? Same question for trees and shrubs, when do you use the basal diameter? How do you distinguish species “autochtonous or not” on the field? How is evaluated the health status? Is it a categorial or a continuous value?

Explanatory variables are neither defined nor described and sourced.

Plant diversity calculations are not clear. For example what is the “Advantage Index” L 140? No explanation or source found. And it is not presented in the results or discussion.
L 161 “ N is the sum of all species, and S is the total number of species” In this sentence, I do not see any difference between N and S.
L 196 “After constructing the model, the variance inflation factor of the model was further checked until all variables had a VIF below 5.” how could you justify this threshold? It is high.
L 201 “corolot function” and “corolot package” are unknow in R.

Validity of the findings

Not checked

---

## Round 0.2 · accepted · Accept

The authors addressed all the comments of the reviewer, and can now be accepted for publication.

·

Basic reporting

No comment.

Experimental design

No comment.

Validity of the findings

No comment.

Additional comments

The authors did a good job of revising the manuscript, improving the English writing, making the manuscript easier to understand.